# Shaping the Innate Immune Response by Dietary Glucans: Any Role in the Control of Cancer?

**DOI:** 10.3390/cancers12010155

**Published:** 2020-01-08

**Authors:** Manuela Del Cornò, Sandra Gessani, Lucia Conti

**Affiliations:** Center for Gender-specific Medicine, Istituto Superiore di Sanità, 00161 Rome, Italy; manuela.delcorno@iss.it (M.D.C.); lucia.conti@iss.it (L.C.)

**Keywords:** innate immunity, β-glucans, nutrition, immunotherapy, cancer

## Abstract

β-glucans represent a heterogeneous group of naturally occurring and biologically active polysaccharides found in many kinds of edible mushrooms, baker’s yeast, cereals and seaweeds, whose health-promoting effects have been known since ancient times. These compounds can be taken orally as food supplements or as part of daily diets, and are safe to use, nonimmunogenic and well tolerated. A main feature of β-glucans is their capacity to function as biological response modifiers, exerting regulatory effects on inflammation and shaping the effector functions of different innate and adaptive immunity cell populations. The potential to interfere with processes involved in the development or control of cancer makes β-glucans interesting candidates as adjuvants in antitumor therapies as well as in cancer prevention strategies. Here, the regulatory effects of dietary β-glucans on human innate immunity cells are reviewed and their potential role in cancer control is discussed.

## 1. Introduction

The immune system has an active role in all phases of carcinogenesis, exerting multifaceted functions that range from antitumoral to protumoral activities. Tumors are populated by a vast and diverse array of immune components: innate sentinels including phagocytes [macrophages, neutrophils and dendritic cells (DC)], natural killer (NK), natural killer T (NKT) and γδ T cells, as well as adaptive leukocytes, including naive, memory, and effector B- and T-lymphocytes [1]. The degree of immune infiltration and the composition of the infiltrate, as well as of the circulating immune cell pool can vary markedly across tumor types and stages [1,2], and can represent important correlates of cancer prognosis and treatment responsiveness [1,2,3]. Although the immune system can elicit an antitumor response that leads to tumor destruction in some cases, the successful development of such a response is often hampered by a plethora of factors. Indeed, immune cell populations are conditioned by soluble factors, enzymes and metabolites produced by nearby tumor and stromal cells within the tumor microenvironment (TME) that contribute to dampen the antitumor immune response [4]. Moreover, the immunosuppressive TME can even shape immune cell functions toward a tumor-promoting response [5]. 

Cancer is frequently associated with chronic inflammation, which is considered a requirement for maintaining an immunosuppressive network. Innate immune cells are the main actors in the inflammatory response by virtue of their expression of pathogen recognition receptors (PRR). The engagement of these receptors not only induces the secretion of proinflammatory cytokines, thereby further supporting inflammation, but also activates antigen presenting cell functions, thus favoring the recognition and killing of tumor cells and the orchestration of a specific adaptive response. 

Shifting the balance towards a protective response has been the goal of many anticancer strategies. An adequate response has been achieved in several human cancers with the use of nonspecific immunotherapies, vaccines, adoptive cell therapy and, more recently, through the blockade of immune checkpoints. Nevertheless, the high antigen-specific T-cell-mediated response measured in some tumor cases does not correlate with clinical benefit [6], suggesting that resistance mechanisms within the TME might orchestrate tumor escape, and that an appropriate activation of innate immunity cells may be the most critical determinant of therapeutic success. 

In recent years, a clear-cut association between nutrition and cancer has been highlighted by several epidemiological and research studies [7]. Dietary patterns and bioactive food-derived compounds have been associated with either increased or decreased risk of several human noncommunicable diseases, including cancer, and their capacity to influence the immune response under physiological and pathological conditions has been widely described (World Cancer Research Fund. Continuous Update Project Expert Report 2018. Diet, Nutrition, Physical Activity and Colorectal Cancer. Available at: dietandcancerreport.org. 2018). There is currently a growing interest in studying the mechanisms underlying the capacity of bioactive food compounds to modulate processes implicated in either the promotion of or in the protection against carcinogenesis. In this regard, the ability of natural food compounds to regulate innate cell functions and to promote or attenuate inflammation is receiving great attention. Among a vast array of compounds naturally occurring polysaccharides, including β-glucans, have been widely studied for their effects on human health and disease in both western and eastern medicine [8,9]. Indeed, despite their structural function, some β-glucans exert important biological activities.

In this review, we will overview the immunomodulatory effects of β-glucans derived from edible mushrooms, baker’s yeast and cereals, assumed as dietary supplements in the form of purified compounds or as extracts. We will focus on human studies and describe the in vitro and in vivo effects of these compounds on innate immunity cells involved in cancer surveillance. The mechanisms underlying the capacity of β-glucans to modulate immune response and how this may impact on tumor development/progression will be discussed.

## 2. Dietary β-glucans: Main Features and Sources

Dietary fibers have been consumed for a long time, due to their beneficial effects on health, as part of the carbohydrate fraction within food [10]. All existing definitions recognize fibers as a group of carbohydrate polymers and oligomers (and lignin) that are resistant to digestion and absorption in the human small intestine where they are, partially or completely, fermented by the gut microbiota. They are present in two main forms depending on their solubility in water (i.e., soluble and insoluble fibers) [8]. Among soluble fibers, β-glucans are homopolysaccharides composed of D-glucopyranosyl residues linked through β-glycosidic bonds comprising a heterogeneous group of naturally occurring biologically active compounds found in the bran of some cereal grains (oat, barley), in the cell wall of baker’s yeast, in many kinds of edible mushrooms (fruit bodies and cultured mycelium) and seaweeds [11,12]. β-glucans of different origins differ in their macromolecular structure and β-linkage, with mushroom- and yeast-derived β-glucans exhibiting primarily β-(1,3/1,6) linkages, while those from cereals have β-(1,3/1,4) linkages [9]. They have the peculiarity of being encountered by humans either as part of their diet or as pathogen associated molecular patterns (PAMP), since they also constitute the cell wall of some pathogenic yeasts and bacteria, and strongly contribute to microorganism recognition and clearance [13]. β-glucans greatly vary in their macromolecular structure, solubility, molecular weight, degree of branching and charge of polymers, as well as in receptor recognition/binding affinity, and their biological activity strongly depends on these features [14,15]. β-glucans derived from different sources appear to show diverse activities. While β-glucans originated from mushrooms are effective as antitumor defense and immunity boost, those found in cereals are more active in lowering cholesterol and blood sugars [15,16]. 

Upon oral ingestion as dietary components, they reach, in an undigested form, the small intestine where intestinal epithelial and/or M cells internalize and present them to the immune cell populations within the Peyer’s patches [17]. β-glucan particles can also reach distant lymphoid organs via blood or lymph [18,19,20]. The recognition by innate immunity cells occurs via ligation of specific PRR, such as Toll-like (TLR) and C-type lectin-like receptors [15,21]. Among the latter, Dectin-1 is the best characterized receptor, reported to bind β-glucan from different sources, and is expressed on the surface of monocytes, macrophages, neutrophils, DC and T lymphocytes [22]. Other PRR, including lactosylceramide receptor, mannose receptor, complement and scavenger receptors were reported to directly bind β-glucan or to cooperate with Dectin-1 for its recognition [15,21,23]. Recently, β-glucan was shown to stimulate NK cell cytotoxic activity through direct binding to the NKp30 activating receptor [24]. Receptor binding on innate immunity cells elicits a number of cellular responses through the modulation of inflammasome and transcription factor activation, and the production of immune response modifiers [cytokines, chemokines, reactive oxygen species (ROS)], finally shaping adaptive immune response.

## 3. Immunomodulatory Effects of β-glucans 

β-glucans from different sources have been extensively studied for their anti-inflammatory, anti-allergic, anti-obesity and anti-osteoporotic activities [9]. The regulatory effects of β-glucans on immune response have been identified as one of the main biological functions that make these compounds attractive targets for therapeutic interventions. Strong evidence on the immunomodulatory activities of β-glucans has been proved by in vitro, as well as by animal- and human-based clinical studies, highlighting their ability to protect against infections and to improve the immunogenicity of vaccines, their antitumor activity and, more recently, their therapeutic potential when combined with other cancer therapies [25,26,27,28]. Indeed some β-glucan preparations have been approved as adjunctive therapeutic drugs for cancer treatment. The safety and lack of toxicity of their oral or intravenous administration have been assayed in phase I–II clinical trials carried out in healthy volunteers, as well as virus-infected or cancer patients [29,30,31,32,33,34]. The size and biochemical composition of β-glucans isolated from different sources have been reported to affect their immunomodulatory properties with the molecular pathways activated and the features of the immune response generated depending on the cell type and receptor triggered [15,35]. Furthermore, β-glucans derived from different sources differ in their capacity to function as PAMP and, as a consequence, in their immunoregulatory efficacy. In this regard, β-glucans derived from baker’s yeast and mushrooms are by far the best-documented immunostimulators, while those from cereals fall short as immune regulators, as they are structurally different and not recognized as PAMP [36,37]. The ensemble of these variables can result in different pharmacologic activities and bioavailability, thus explaining the discrepancies observed when comparing different studies. 

The main evidence on the in vitro effects of β-glucans on human innate immunity cells, as well as the clinical studies assessing their in vivo effects on the same cell populations in either healthy subjects or cancer patients, are reviewed in the next sections and summarized in Table 1, Table 2 and Table 3. 

### 3.1. In Vitro Effects on Innate Immunity Cells 

In vitro studies have shown that β-glucans from yeasts, mushrooms or cereals are able to enhance the responsiveness or function of human primary immune cells, eliciting potent immune responses through their recognition by a variety of PRR, particularly Dectin-1 and complement receptor 3 (CR3). Within the innate immune system, the targeted cells of β-glucans include macrophages, neutrophils, monocytes, NK cells and DC. Specifically, β-glucans can enhance the functional activity of monocytes/macrophages and DC and activate antimicrobial activity of mononuclear cells and neutrophils in vitro. This enhanced immune response is accomplished by an increased proinflammatory cytokine and chemokine production and an enhanced oxidative burst, as mainly demonstrated in mouse models [38].

**Table 1 cancers-12-00155-t001:** In vitro studies evaluating the immunomodulatory effects of β-glucans on human innate immunity cells.

Compound (Concentration Range ^1^)	Cell Type	Effects	Molecular Mechanisms	Refs
*Agaricus brasiliensis* acid-treated polysaccharide-rich fraction (50 µg/mL)	Monocytes	↑ Adherence ↑ Phagocytosis ↑ TNFα , IL-1β, IL-10	↑ TLR2 and TLR4 = βGR or MR	[39]
*Flammulina velutipes* extract	Monocytes Macrophages	↑ Cytokine production ↑ Phagocytosis ↑ ROS		[40]
*Agaricus blazei* Murill extract (1–15%)	Monocytes	↑ IL-8, TNFα, IL-1β, IL-6		[41]
*Agaricus blazei* Murill extract (0.5–15%)	Monocytes	↑ Phagocytosis	↑ CD11b ↓ CD62L	[42]
*Pleurotus citrinopileatus* polysaccharide (PCPS, 0.5 µg/mL)	Monocytes Macrophages	Differentiation of monocytes toward macrophages (IFNγ + LPS) with reduced proinflammatory capacity: ↓ TNFα, IL-6 and CCR2 mRNA ↑ IL-10, CCL2 and CCL8 mRNA	Dectin-1 and TLR2 signaling pathways	[43]
*Piptoporus betulinus* extract	Monocytes	↓ Apoptosis ↑ IL-8		[44]
	MDDC	↑ Maturation ↑ IL-8		
*Pleurotus citrinopileatus* polysaccharide (PCPS, 0.01–5 µg/mL)	MDDC	↑ CD80, CD86, HLA-DR ↑ Pro- and anti-inflammatory cytokines (TNFα, IL-1β, IL-6, IL-12, IL-10) ↑ mRNA: CCL2, CCL3, CCL8, CXCL9, CXCL10, and LTA	Dectin-1, TLR2 and TLR4 signaling pathways	[45]
*Armillariella mellea* water-soluble components (2–20 µg/mL)	MDDC	↑ CD80, CD83, CD86, MHC class I and II, CD205 ↓ CD206 ↓ Endocytic capacity = TNFα, IL-12, IL-10		[46]
*Hericium erinaceum* water-soluble components (2–20 µg/mL)	MDDC	↑ CD80, CD83, CD86, MHC class I and II, CD205 ↓ CD206 ↓ Endocytic capacity = TNFα, IL-12		[47]
*Agaricus blazei* Murill Extract (10% ABM = 2.8 g of β-glucan/100 g)	MDDC	↑ IL-8, G-CSF, TNFα, IL-1β, IL-6, CCL4		[48]
Various higher Basidiomycetes exctracts (0.0005–5 mg/mL)	Neutrophils	↑ ROS		[49]
*Pleurotus ostreatus* β-glucans extracted from fruiting bodies (5 mg/mL)	NK	↑ Cytotoxic effects against lung and breast cancer cell lines	↑ KIR2DL genes ↑ NKG2D, IFNγ, NO	[50]
*Grifola frondosa* polysaccharide (10 mg/L), Lentinan (500 mg/L), Yeast glucan (100 mg/L)	NK	↑ Cytotoxicity, IFNγ, perforin secretion	↑ NKp30 expression	[51]
*Saccaromyces cerevisiae* glucan from baker’s yeast and zymosan (10 or 100 µg/mL)	Macrophages	↑ IL-1β transcription and secretion	Dectin 1/Syk signaling pathway NLPR3 activation	[52]
*Saccaromyces cerevisiae* whole β-glucan particles (100 µg/mL)	MDDC	↑ CD40, CD86, HLA-DR ↑ IL12, IL-2, TNF, IFNγ ↑ CD8 T cell priming ↑ Tumor-specific CTL activity	PI3K/Akt signalling	[53]
*Saccaromyces cerevisiae* baker’s yeast	MDDC	↑ Th17 cells ↑ Adhesion and migration	IL-1α, IFNγ	[54]
*Saccharomyces cerevisiae* zymosan (1 mg/mL)	MDDC	↑ IL-23	LTB4, PAF	[55]
*Saccharomyces cerevisiae* zymosan (1 mg/mL)	MDDC Macrophages	↑ p-STAT3 ↑ mRNA: IL-10, IL-23, INF1B, CSF1, CSF2 and CSF3	PGE2	[56]
*Saccaromyces cerevisiae* Imprime PGG (10 µg/mL)	Monocytes	↑ ADCP ↑ C5a ↑ IL-8, CCL2 ↑ CD11b ↓ CD62L, CD88, CXCR2 ↑ Phenotypic and functional activation	Formation of an immune complex with naturally occurring ABA	[57]
	Neutrophils	↑ ROS ↑ IL-8, CCL2 ↑ CD11b ↓ CD62L, CD88, CXCR2		
*Saccaromyces cerevisiae* Betafectin PGG (0–300 µg/mL)	Neutrophils	↑ Chemotaxis toward C5a ↓ Chemotaxis toward IL-8	CR3-dependent	[58]
Barley polysaccharides (100 µg/mL)	MDDC	↑ Phenotypic and functional maturation of DC ↑ IL-12, IL-10		[59]
Barley β-glucan	Umbilical cord blood-generated DC	↓ CCL2 ↑ CD83 cells		[60]

Abbreviations: anti-β glucan antibodies, ABA; antibody-dependent cellular phagocytosis, ADCP; β-glucan receptor, βGR; cytotoxic T lymphocyte, CTL; dendritic cells, DC; Killer immunoglobulin receptor, KIR; Lymphotoxin a, LTA; leukotriene B4, LTB4; monocyte-derived dendritic cells, MDDC; mannose receptor, MR; natural killer, NK; nitric oxide, NO; platelet-activating factor, PAF; prostaglandin E2, PGE2; reactive oxygen species, ROS. ^1^ β-glucan concentrations are shown when available.

As shown in Table 1, biologically active fungal β-glucans found in edible mushrooms (e.g., *Agaricus brasiliensis*, *Pleurotus citrinopileatus, Agaricus blazei*, *Flammulina velutipes*) can enhance inflammatory cytokine production in monocytes [39,40,41,43,44] as well as in monocyte-derived macrophages (MDM) [40]. Furthermore, Minato and colleagues have recently explored the functional effect of *Pleurotus citrinopileatus* polysaccharide (PCPS) on monocyte-to-macrophage differentiation, finding that PCPS can direct Dectin-1 and TLR2-mediated differentiation of monocytes toward a macrophage cell population with reduced proinflammatory capacity [43]. Similarly, the function of DC can be influenced by β-glucans from mushroom extracts in terms of secretion of proinflammatory cytokines/chemokines [45,48], through Dectin-1 signaling and Syk/Raf-1-dependent pathways [45]. In addition, it has been reported that β-glucans derived from *Armillariella mellea*, *Hericium erinaceum* and *Piptoporus betulinus*, enhance phenotypic and functional maturation of monocyte derived dendritic cells (MDDC), with significant interleukin (IL)-12 and IL-10 production, and upregulation of surface maturation markers [44,45,46,47,48]. In general, β-glucans of a larger size and more branching complexity have higher immunomodulating potency [59], while the effect of fruit body extracts of various higher Basidiomycetes are slightly higher than those observed for mycelium extracts of the same species [49]. 

Additional studies indicated that also β-glucans isolated from yeasts overall enhance the immune response by modulating proinflammatory cytokine production in macrophages via Dectin-1/Syk signaling pathway and NLPR3 inflammasome activation, as well as in MDDC via PI3K/AKT signaling pathway [52,53,54]. Moreover, yeast-derived β-glucans induce DC maturation, significantly increase tumor-specific cytotoxic T lymphocyte (CTL) responses [53] and program DC to express cell adhesion and migration mediators, antimicrobial molecules, and Th17-polarizing factors [54]. Furthermore, the role of secondary mediators, which include lipids, in the induction of the cytokine signature was demonstrated in DC stimulated with zymosan, a cell-wall extract from *Saccharomyces cerevisiae* that contains a β-glucan component, providing mechanistic clues that β-glucans can modulate the immune response by acting on the lipid mediator cascade and by activating STAT3 signaling [55,56]. The regulation of additional cellular responses by mushroom-derived β-glucans in monocytes and DC has been also described. Specifically, β-glucans from *Agaricus brasiliensis*, *Agaricus blazei* or *Flammulina velutipes* extracts with immunomodulatory and antitumor activities, significantly increase the adherence and phagocytosis by monocytes [39,40,41] and enhance ROS production in monocytes and macrophages [40], while water-soluble components from *Armillariella mellea* or *Hericium erinaceum* decrease the MDDC endocytic capacity [46,47]. Besides β-glucans from yeasts and mushrooms, also those derived from cereals influence MDDC functions. The interaction of barley β-glucan with DC enhances maturation and cytokine production, leading to changes in the balance of Th1/Th2 cytokines in autologous T cells [60]. However, the effects of this compound on DC maturation/activation are rather weak as compared to those of β-glucans derived from mushrooms [59]. 

Large β-glucans can be degraded by macrophages into smaller fractions that, when released, prime CR3 on neutrophils and NK cells, triggering antimicrobial activity. Indeed, poly-[1–6]-d-glucopyranosyl-[1–3]-d-glucopyranose (PGG) glucan from yeast *Saccharomyces cerevisiae* (Imprime PGG) as well as aqueous extracts from fruit bodies and mycelia of various higher Basidiomycetes, enhance ROS production and antibody-dependent cellular phagocytosis by neutrophils [49,57,61] and monocytes [57], respectively. Mechanistically, Imprime binding and functional activation of these innate effector cells require first the formation of an immune complex with naturally occurring anti-β-glucan antibodies and subsequent opsonization by complement proteins [57,62], the activation of NF-kB [61], which results in enhanced cell migration toward the chemoattractant C5a [58]. Conversely, studies investigating the biological effects of β-glucans on NK cells have been rarely performed. Nevertheless, the effects of fungal- or yeast-derived β-glucans on primary NK cells demonstrate that these polysaccharides markedly enhance NK cell cytotoxicity by stimulating interferon (IFN)γ and perforin secretion and increasing the expression of the activating receptor NKp30, with CR3 as a critical receptor [51]. Moreover, β-glucans present in *Pleurotus ostreatus* extracts have the ability to induce NK-cell mediated cytotoxicity against cancer cells and to enhance cytokine production. The cytotoxic effects are mediated by NKG2D and IFNγ upregulation and are enhanced in the presence of IL-2, while the activation for cytokine secretion is associated with upregulation of KIR2DL genes [50]. 

In summary, these studies suggest that β-glucans, derived from edible mushrooms, yeasts or cereals, have the potential to regulate a plethora of immune functions. β-glucans binding to specific receptors on DC and macrophages trigger their activation and maturation, enhance the production of proinflammatory cytokines that in turn stimulate the polarization toward Th1 or Th17 responses, and induce the activation of antigen-specific CD8^+^ CTL. Furthermore, β-glucans can enhance the antimicrobial activity of neutrophils and promote NK cell cytotoxicity.

### 3.2. In Vivo Effects in Healthy Subjects

A number of human studies involving healthy volunteers have examined the effects of dietary supplementation with β-glucans of different origins on innate immune responses (Table 2). 

**Table 2 cancers-12-00155-t002:** Main clinical studies evaluating the immunomodulatory effects of β-glucans in healthy subjects.

Compound (Concentration Range ^1^)	Subjects/Study Type	Control Group	β-glucan Group	Major Findings	Refs.
*Pleurotus ostreatus* β-glucan (Pleuran, 100 mg/day)	Regularly training athletes/Randomized	Vitamin C (*n* = 25)	β-glucan + vitamin C (*n* = 25)	↑ NK cell frequency ↑ PMN-mediated phagocytosis ↓ URTI symptom incidence	[63]
*Pleurotus ostreatus* β-glucan (Pleuran, 100 mg/day)	Elite athletes/Randomized	Fructose + vitamin C (*n* = 11)	β-glucan + vitamin C (*n* = 9)	Restrained high intensity PA-induced reduction of NK cell number and activity = Monocyte and granulocyte counts	[64]
*Pleurotus cornucopiae* water extract (β-glucan 24 mg/meal)	Healthy volunteers/ Randomized	Water, tea, oyster souce, caffeine-free coffee (*n* = 21)	β-glucan + water, tea, oyster souce, caffeine-free coffee (*n* = 20)	↑ NK cell activity ↑ Th1-type response	[65]
*Agaricus blazei* Murill extract (AndoSan, 60 mL/day = 5.7 g β-glucan)	Healthy volunteers/Intervention study	None	Mushroom extract (*n* = 10)	↓ Intracellular ROS in monocytes and granulocytes vs baseline	[66]
*Grifola frondosa* extract (6 mg/kg/day)	Myelodysplastic syndrome patients/ Non randomized phase II trial	None	Mushroom extract (*n* = 21)	↑ Neutrophil and monocyte functions (ROS production)	[67]
Oat soluble β-glucan (5.6 g)	Trained male cyclists (on intense exercise)/ Randomized	Cornstarch + Gatorade (*n* = 20)	β-glucan (Oatvantge) + Gatorade (*n* = 20)	No rescue of NK cell activity No rescue of PNM-RBA No effect on URTI symptom incidence	[68]
*Saccharomyces cerevisiae* β-glucan (Purified, Imuneks, 20 mg/day)	Subjects with seasonal allergic rhinitis (allergen sensitized)/ Randomized	Nihil (*n* = 12)	β-glucan (*n* = 12)	↓ Eosinophil frequency in the nasal fluid lavage	[69]
*Saccharomyces cerevisiae* β-glucan (insoluble, 1 g/day)	Healthy volunteers/ Intervention study	Nihil (*n* = 5)	β-glucan (*n* = 10)	No effect on phagocyte functions (cytokine production + microbicide activity)	[70]

Abbreviations: Natural Killer, NK; Physical activity, PA; Polymorphonuclear leukocyte, PMN; Reactive oxygen species, ROS; Respiratory burst activity, RBA; Subjects’ number, N; T helper, Th; Upper respiratory tract infection, URTI. ^1^ β-glucan concentrations are shown when available.

In keeping with the in vitro studies, these compounds were reported to affect either the abundance/frequency or the effector functions of different innate cell populations. In a study performed on regularly training athletes (from different sport disciplines) receiving a three months supplementation with insoluble, particulate β-glucan (Pleuran, Imunoglukan) extracted from the fruit bodies of *Pleurotus ostreatus*, the number of circulating NK cells significantly increased in the treated as compared to the placebo group [63]. In agreement with data from in vitro studies [49], a concomitant enhancement of polymorphonuclear leukocyte (PMN)-mediated phagocytosis was observed in treated subjects, together with a decreased incidence of upper respiratory tract infection (URTI) symptoms [63]. Moreover, a two-month oral supplementation with the same compound (Pleuran) was found to counteract the decrease of blood NK cell number and activity (NKCA), but not that of granulocytes and monocytes, induced by short-term high-intensity exercise in elite athletes [64]. This evidence points to a role of *Pleurotus ostreatus* insoluble β-glucan not only as immune modifier and adjuvant of regular physical activity but also in the prevention of immunosuppression induced by acute exhausting physical load. Moreover, these studies, together with in vitro evidence on the enhancement of NK cell cytotoxicity against tumor cells [50], highlight the peculiar feature of this compound to boost innate immunity mainly by acting on NK cells. β-glucan-mediated increase of NKCA and Th1-type immune responses, although not statistically significant, was also observed in a Japanese clinical trial in which the subjects ingested water extracts (β-glucan content 24 mg/meal) from Oyster mushroom Pleurotus *Cornucopiae* (Tomogitake) for 8 weeks [65]. In contrast, a beverage supplement containing soluble β-glucan extracted from oat did rescue neither NKCA and PMN respiratory burst activity, nor the incidence of URTI symptoms, with respect to a placebo over an 18-day intake in trained male cyclists [68,71].

Regulatory effects on blood granulocytes and monocytes (i.e., increased frequency or reduction of proinflammatory cytokine and ROS production) have been described after oral administration of extracts from the higher Basidiomycetes *Grifola frondosa* and *Agaricus blazei* Murill (AbM, AndoSan) [66,67]. However, although these extracts are characterized by a high content of β-glucan, the specific contribution of the glucan fraction to the immunomodulatory activity has not been demonstrated in these studies [66,67]. Finally, β-glucan purified from the cell wall of *Saccharomyces cerevisiae* (Imuneks) was reported to reduce some immunopathogenic processes characterizing subjects with allergic rhinitis [69]. In particular, a 12-week oral supplementation with β-glucan in allergen-sensitized subjects resulted in a significant decrease in the frequency of eosinophils, crucial effectors in allergic reactions, in the nasal lavage fluid [69]. Conversely, in a pilot intervention study on healthy volunteers, dietary supplementation with *Saccharomyces cerevisiae* insoluble β-glucan for 7 days did not affect cytokine production and microbicide activity of peripheral blood phagocytes [70]. 

With the exception of some studies on animal models, the impact of dietary whole mushroom consumption on immunity has been only poorly investigated in humans. In a study by Dai and colleagues, a regular consumption of dried *Lentinula edodes* mushrooms (5 or 10 g/day) for four weeks was found to significantly increase the activation and proliferative potential of innate lymphocytes, particularly γδ T and NKT cells, concomitantly with a decrease of systemic inflammatory markers [72]. Regardless of the mechanism(s) and bioactive fraction(s) responsible for this activity, these results suggest that regular dietary intake of mushrooms could improve innate immunity and dampen inflammation.

### 3.3. In Vivo Effects in Cancer Patients

The effects of dietary supplementation with β-glucans of different origins have been investigated in a number of clinical trials carried out in cancer patients either undergoing conventional anticancer therapy, or in the absence of any other concomitant therapeutic option (Table 3).

**Table 3 cancers-12-00155-t003:** Main clinical studies evaluating the immunomodulatory effects of β-glucans in cancer patients.

Compound (Concentration Range ^1^)	Cancer Type	Conventional Therapy	Treated Patients (*N*)	Major Findings	Refs.
*Agaricus blazei* Murill extract	Gynecological	Yes	39	↑ NK cell activity = LAK and monocyte activity ↓ Chemotherapy-induced side effects	[73]
*Agaricus blazei* Murill extract (AndoSan, 60 mL/day = 5.7 g β-glucan)	Multiple myeloma	Yes	19	↑ T_reg_ and pDC numbers ↑ IL-1Ra, IL-5, IL-7 ↑ Ig, KIR, HLA gene expression	[74]
*Lentinula edodes* mycelia extract (1.8 g/day)	Advanced breast	Yes	10	Restrained chemotherapy-induced reduction of NK and LAK cell activity and of white blood cell/neutrophil counts ↑ QOL	[75]
*Lentinula edodes* mycelia extract (1.8 g/day)	Breast, gastric, colorectal, esophageal	Yes	7	↑ NK cell and LAK activity ↑ QOL ↓ IAP	[76]
*Lentinula edodes* β-glucan Lentinan (1 mg/every other day)	Esophageal	Yes	25	↓ Chemotherapy side effects ↑ QOL ↑ IL-12, IL-2, IL-6 ↓IL-4, IL-5, IL-10	[77]
*Lentinula edodes* β-glucan Lentinan	Gastric	Yes	20	↑ QOL	[78]
*Lentinula edodes* β-glucan Lentinan (2 mg/Kg/week)	Unresectable or recurrent gastric	Yes	147	= Leukocyte and neutrophil counts = Side-effects = QOL	[79]
Yeast β-glucan (Purified, Imuneks, 20 mg/day)	Advanced breast	Yes	15	Restrained chemotherapy-induced reduction of white blood cells = Neutrophil and monocyte counts ↑ IL-12 ↓IL-4 ↑ QOL	[80,81]
Yeast β-glucan (Purified, Imuneks, 20 mg/day)	Advanced breast	Yes	8	↑ CD14^+^ monocyte number ↑ CD95 and CD45RA expression in monocytes	[82]
*Agaricus bisporus* powder (4–14 g/day)	Recurrent prostate	No	36	↓ MDSC numbers ↑ IL-15	[83]
*Grifola frondosa* D-Fraction (40–150 mg/day)	Advanced lung and breast	No	10	↑ NK cell activity	[84]
*Grifola frondosa* D-Fraction (0.1–5 mg/twice/day	Breast	No	34	↑ NKT and T_reg_ cell numbers ↑ Response of ex-vivo immune cells	[85]
Yeast β-glucan (500 mg/day)	Newly diagnosed NSCLC	No	23	↓ MDSC numbers	[86]

Abbreviations: Patients’ number, N; Immunoglobulin, Ig; Immunosuppressive acidic protein, IAP; Killer immunoglobulin receptor, KIR; Lymphokine-activated killer, LAK; Myeloid-derived suppressor cell, MDSC; Natural killer, NK; Non-small cell lung carcinoma, NSCLC; Plasmacytoid DC, pDC; Quality of life, QOL; Regulatory T, T_reg_. ^1^ β-glucan concentrations are shown when available.

The majority of these studies investigated changes in patients’ immune system by monitoring surrogate markers such as leukocyte counts or cytokine profiles as well as by measuring chemotherapy-associated side effects and indexes of quality of life (QOL). One of the most prominent effect described for the oral administration of β-glucans is the attenuation of chemotherapy-induced fall of leukocyte counts paralleling numerical or functional changes of some specific immune cell populations. As shown in Table 3, *Lentinula edodes* extracts and yeast-derived β-glucan (Imuneks) counteract the chemotherapy induced decrease in NK and lymphokine-activated killer (LAK) cell activity when administered to advanced breast cancer patients [75,80]. Likewise, NK, NKT and LAK cell cytotoxic activity was reported to be higher in cancer patients administered with *Agaricus blazei* or *Lentinula edodes* extracts [73,75,76]. Conversely, the administration of *Agaricus blazei* Murill extracts (AndoSan) to multiple myeloma patients undergoing high dose chemotherapy resulted in the expansion of different immune cell populations such as regulatory T (T_reg_) cells and plasmacytoid DC, and in enhanced serum levels of IL-Ra, IL-5 and IL-7. Furthermore, the expression of immunoglobulin and NK cell-related genes was also observed [74]. The capacity of β-glucans to numerically/functionally influence specific subsets of immune cells has been also documented by studies assessing the effects of yeast-derived β-glucan (Imuneks) in patients affected by advanced breast cancer. These studies reported that β-glucan stimulates the proliferation and activation of CD14^+^ monocytes expressing high levels of CD95 and CD45RA surface antigens [82]. 

The ability of β-glucan to counteract, at least in part, the negative effect of chemotherapy on blood cell counts mainly relies on its hematopoiesis-enhancing capacity. In this regard, administration of carboxymethyl-glucan (CM-G), a water-soluble derivative of *Saccharomyces cerevisiae*, to advanced prostate cancer patients, has been reported to enhance hematopoietic regeneration. After CM-G administration, the total leukocyte, red blood cell and platelet counts, as well as hematocrit and hemoglobin increased significantly independently of changes in the lifestyle habits of patients [87].

Numerical and functional changes of immune cell populations have been also observed following β-glucan supplementation to cancer patients in the absence of any other therapeutic option, either in newly diagnosed patients or in those who have already completed their therapeutic pathway. In this regard, in non small cell lung carcinoma patients, the administration of yeast-derived particulate β-glucan soon after diagnosis decreases the frequency of circulating CD33^+^HLA-DR^−^ myeloid-derived suppressor cells (MDSC) with improved effector function [86]. A similar decrease in the percentage of CD33^+^HLA-DR^−^ MDSC was also observed in a phase I trial involving patients with biochemically recurrent prostate cancer following supplementation with *Agaricus bisporus* powder [83]. Other clinical studies highlighted the effects of the administration of β-glucan enriched D-Fraction, extracted from the fruit body of the Maitake mushroom *Grifola frondosa*, to either advanced lung or breast cancer patients [84] as well as to breast cancer patients free of disease after initial treatments [85]. While in patients at advanced stages of disease the administration of D-fraction promoted NK cell activation [84], in those free of disease the main changes were observed at the level of CD3^+^ CD56^+^ NKT and CD4^+^ CD25^+^ T_reg_ cells, whose frequency was increased in the β-glucan treated group [85].

Some studies also reported modulations of cytokine serum levels as well as functional changes of ex-vivo immune cell populations upon in vitro stimulation. In particular, increased secretion of IL-12 paralleling a decrease in IL-4 content was observed in advanced breast cancer patients administered with Imuneks [80] as well as in esophageal cancer patients supplemented with Lentinan, a purified β-glucan from *Lentinula edodes* [77]. Interestingly, a significant increase in the serum content of IL-15, a cytokine promoting NK cell differentiation and optimizing NK cell function, was detected upon supplementation with *Agaricus bisporus* [83]. Likewise, functional changes were observed in ex-vivo immune cells from breast cancer patients free of disease including granulocyte response to phorbol myristate acetate (PMA) stimulation, IL-10 production by PMA stimulated CD14^+^ cells, and constitutive IL-2 secretion by CD3^+^ CD56^+^ NKT cells [85]. 

The most promising evidence to date in clinical trials has come from studies investigating the benefits of β-glucan, in combination with conventional cancer treatment, on therapy side effects, QOL and survival of cancer patients. Indeed, a significant reduction of chemotherapy-associated side effects such as loss of appetite, alopecia, emotional instability, and general weakness was reported in β-glucan supplemented patients, leading to a general improvement of QOL [73,75,76,77,78,81]. In this regard, a recent systematic review of clinical applications of Lentinan in treating lung cancer during the past 12 years in China reported a solid effect of Lentinan on improving QOL and on promoting the efficacy of chemotherapy and radiation therapy [88]. In contrast with these results, Yoshino and colleagues failed to detect any change in leukocyte/neutrophil counts or any improvement in chemotherapy-induced side effects and QOL in unresectable or recurrent gastric cancer patients upon administration of Lentinan. These discrepancies likely rely not only on differences in the compound administered but also in the type of cancer and related chemotherapeutic options. However, it is of interest that a subpopulation of patients with Lentinan-binding monocytes ≥ 2% in the peripheral blood showed a longer survival time when this compound was administered together with chemotherapy, suggesting that Lentinan-binding monocytes could represent a potential biomarker predicting chemotherapy response [79].

Taken together, these observations suggest that β-glucan provides an additional alternative therapeutic modality to maintain innate immunity cell activity and, in particular, to reduce many severe side effects caused by chemotherapy in cancer patients by stimulating hematopoiesis and assisting with an overall improvement in quality of life. 

## 4. Conclusions

Nowadays, dietary β-glucans consumed in a daily diet or as a food supplement are well studied for their multifaceted activities with important implications for human health. Multiple effects have been reported for these compounds, including metabolic and anti-obesity activities, anti-osteoporotic, anti-allergic and anti-inflammatory effects, promotion of hematopoiesis, and regulation of gut commensal flora (Figure 1). The anticarcinogenic effects of β-glucans are clearly evidenced in the scientific literature and include: (i) the direct control of cancer cell growth, (ii) the modulation of TME, both by bridging the innate and adaptive arms of the immune system and by orienting immunosuppressive cells toward an immunostimulatory phenotype, (iii) their synergism with conventional anticancer therapy. In this regard, preclinical studies have demonstrated that the administration of certain β-glucans can effectively manipulate TME, for example by inducing the phenotypic conversion of tumor associated macrophages, leading to a significant reduction of primary tumor and distant metastases [89,90]. 

In this review, we focused on the immunomodulatory effects of β-glucans indirectly affecting tumor development/progression that may be relevant for their potential application in the fight against cancer in combination therapies. As schematically represented in Figure 1, multiple innate immunity cell subsets and biological processes are modulated by β-glucans, ranging from secretion of immune mediators to induction of phagocytosis and cytotoxic functions. Furthermore, the capacity of these compounds to attenuate chemotherapy-induced side effects, to promote hematopoiesis and to improve QOL has been reported in cancer patients. The ensemble of effects on innate immunity cells, key players in cancer surveillance, observed in vitro as well as in preclinical and clinical studies, clearly highlights the potential of β-glucans to modulate key processes involved in carcinogenesis and immune-mediated control of cancer, and make these compounds attractive as adjuvants in cancer therapies. 

Nevertheless, the emerging success of combinatorial approaches to cancer treatment involving Imprime, a yeast-derived soluble β-glucan in clinical development for cancer, as an intravenously administered immunotherapy, in combination with immune checkpoint inhibitors (CPI), expanded the clinical benefit of CPI therapy by enhancing the anticancer immune response. The combinatorial use of β-glucan with immunotherapy or conventional cancer therapy is beginning to show great promise in improving patient morbidity and mortality, strongly suggesting that β-glucans may play an essential role in future strategies to inhibit tumor growth and to improve the clinical responses to current therapies. To this aim, studies that deeply characterize how β-glucans with different structures and size interact with specific receptors and regulated molecular pathways—as well as the use of β-glucan molecules with precisely defined biochemical properties—are necessary for the full exploitation of these food-derived biological response modifiers in the fight against cancer.

## Figures and Tables

**Figure 1 cancers-12-00155-f001:**
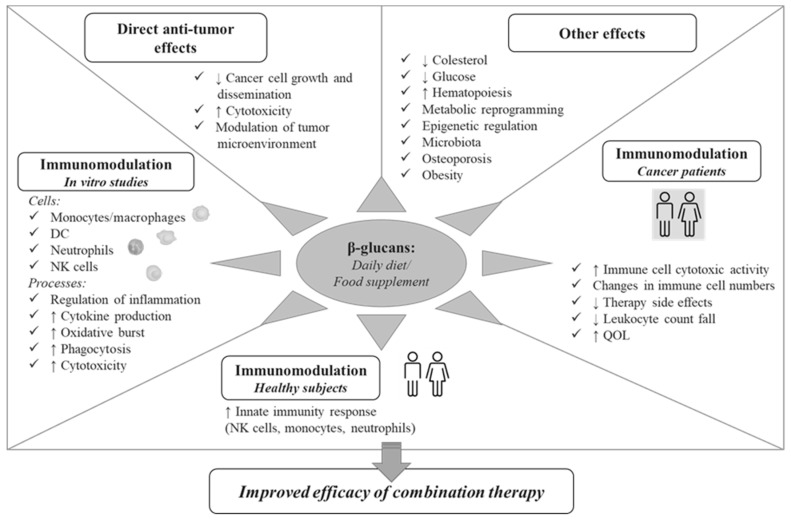
A schematic representation of the potential role of β-glucan in the control of cancer.

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
