# Peer review of "Shaping the Innate Immune Response by Dietary Glucans: Any Role in the Control of Cancer?"

_cancers, 2020, doi:10.3390/cancers12010155_

Round 1

Reviewer 1 Report

This is a nicely written review article centering on the effect of dietary glucans in the control of cancer. The authors have mainly discussed its role in shaping the innate immune response. There are some minor comments on this review manuscript.

Macrophages are known to be stimulated by dietary glucans to produce proinflammatory cytokines. In addition, yeast-derived particulate beta-glucan has been shown to convert immunosuppressive macrophages into antitumor M1-like phenotype (PMID:26453753). One of the challenges in the glucan field is related to the dose and purify of different sources of glucans. The authors may wish to elaborate more on this topic. In addition, if these information is available, authors should include them in the Tables. Without these information, it is hard to compare the effect of different glucans.  Similarly, the size and the structure of different dietary glucans also impact on their effects. Please include these information (if they are available) when these glucans are described.

Author Response

Macrophages are known to be stimulated by dietary glucans to produce proinflammatory cytokines. In addition, yeast-derived particulate beta-glucan has been shown to convert immunosuppressive macrophages into antitumor M1-like phenotype (PMID:26453753).

As suggested by the Reviewer, this information has been added in the Conclusions Section (new reference 90).

One of the challenges in the glucan field is related to the dose and purify of different sources of glucans. The authors may wish to elaborate more on this topic. In addition, if these information is available, authors should include them in the Tables. Without these information, it is hard to compare the effect of different glucans.  Similarly, the size and the structure of different dietary glucans also impact on their effects. Please include these information (if they are available) when these glucans are described. 

We agree with the Reviewer that significant challenges for β-glucan research and clinical application are the high structural variability, the different origin and degree of purity, and the promiscuity of receptors engaged, the purity and dose as we discussed throughout the text (Section 3, lines 112-114; Discussion, lines 360-367). Although it is not always possible to exactly establish the β-glucan concentration in the polysaccharidic fraction of the different preparations used, with the exception of the β-glucan purified preparations, we implemented all tables by providing whenever indicated, the β-glucan concentration. Further information on the molecular structure and β linkages of β-glucans derived from the 3 major sources described in this review: edible mushrooms, yeasts and cereals (lines 78-80).

Reviewer 2 Report

In this review, the regulatory effects of dietary β-glucans on human innate immunity cells and their potential role in cancer control are discussed. The review is outstanding, it is written very well and it covers comprehensively the topic. I congratulate with the authors.

I have only one comment: the Review would benefit from a Table illustrating which kind of glucans are contained in which food type, and relative abundance. Perhaps indicating if glucans are more represented in vegan/vegetarian versus Western diets. This information is partially contained in the text, but it would be nice and timely to show for the readers.

Author Response

The Review would benefit from a Table illustrating which kind of glucans are contained in which food type, and relative abundance. Perhaps indicating if glucans are more represented in vegan/vegetarian versus Western diets. This information is partially contained in the text, but it would be nice and timely to show for the readers.

We have taken into consideration the Reviewer’s suggestion but since the review has already 3 rather large tables, we preferred to implement the information contained in the text rather than adding a new Table, better describing the differences in structure and β linkages of β-glucans derived from the 3 major sources described in this review: edible mushrooms, yeasts and cereals. It is reasonable to assume that the intake of β-glucans in vegetarian/vegan diets is superior to that in Western diet due to the higher consumption of fibers and cereals. However, there is no available literature specifically addressing this issue.